**Data Availability Statement:** All relevant data are within the paper and its Supporting Information files.

**Funding:** This research was funded by National Social Science Foundation of China, grant number

# Does Tobler's first law of geography apply to internet attention? A case study of the Asian elephant northern migration event

**Boming Zheng[1], Xijie Lin[1], Duo Yin[2], Xinhua Qi[1,3,4]***

**1** School of Geographical Sciences, Fujian Normal University, Fuzhou, China, **2** School of Geography and Remote Sensing, Guangzhou University, Guangzhou, China, **3** Institute of Geography, Fujian Normal University, Fuzhou, China, **4** Key Laboratory for Humid Subtropical Eco-Geographical Processes of the Ministry, Fujian Normal University, Fuzhou, China

\* fjqxh74@163.com

## Abstract

One of the basic assumptions of spatial theory is formulated in Waldo Tobler's first law of geography: "everything is related to everything else, but near things are more related than distant things." However, as internet space is a complex virtual space independent of the real world, whether this law is applicable to things in the internet space remains to be explored in depth. Therefore, this study takes the event of Asian elephant northern migration as an example, attempts to investigate the issue of the applicability of Tobler's first law of geography to internet attention by integrating geographic methods such as spatial visualization, spatial correlation analysis, and Geo-detector. The results show that Tobler's first law of geography does not fully apply to internet attention, which does not decay with increasing distance. Geographical distance, within certain boundaries, is influenced by "identity" and "relevance", and still plays a large role in internet attention. However, once the boundaries are exceeded, the impact of geographic distance on internet attention is weakened by the intervention of influencing factors such as the degree of information technology, population, and the strength of news media publicity. Overall, the strength of news media publicity has the greatest impact on internet attention. And when it interacts with geographic proximity, it has the most significant effect on internet attention.

## Introduction

Since the 1990s, human society has gradually moved from the industrial age to the information age [1]. The rapid spread of the internet makes information dissemination no longer limited by time and space, and extensive information is spreading rapidly and instantly. The China Internet Development Report (2021) shows that the number of Chinese Internet users reached 989 million by the end of 2020, and access to all kinds of information has become increasingly prevalent. Traditional geographical laws such as "distance decay" and "diminishing benefits" are defined in this context as "spatial and temporal compression" [2], and pessimistic arguments including "death of distance" [3] and "the end of geography" [4] frequently emerge in

18BJL126. National Natural Science Foundation of China, grant number 41901173. The funders' contributions to the paper were mainly in supervision, funding acquisition, and project administration.

**Competing interests:** The authors have declared that no competing interests exist.

academic circles. To respond to the challenges of the information age and explore the laws of space and time, the research framework of geography has been widened into sub-disciplines, including information geography [5], communication geography [6,7], telecommunication geography [8], cyberspace geography [9], virtual geography [10], and media geography [11]. The aim is to explore the characteristics and laws of the territorial system of human-land relations from a geographic perspective in the information age, thereby expanding the research scope of geography [12]. In this context, this study attempts to investigate the applicability of Tobler's first law of geography to internet attention (An indicator to measure the active search behavior of internet users based on their subjective information needs, which characterizes the attention orientation of the public) in the hope of enriching and improving the theoretical perception of distance decay in the internet era.

Tobler's first law of geography states that everything is related to everything else, but near things are more related to each other [13]. This law mainly explains the frictional effect of distance on spatial interactions and considers that the strength of spatial interactions decreases as the distance between locations increases [14]. In the real world, many things and activities exist in a specific macro or micro space, and will undergo some spatial variation with the migration of time. Because of this, Tobler's first law of geography has long had a significant impact on many natural and humanities fields such as ecology [15–17], biology [17,18], tourism [19,20], transportation [21,22], and behavior [23]. Bjorholm took American palms (Arecaceae) as an example, assess the extent to which Tobler's first law applies to species richness and species composition. To shed light on the mechanisms driving distance decays in community structure and quantify the relative contribution of geographic distance per se and environmental changes as drivers of spatial turnover in species richness and composition [15]. Siewert demonstrated the great spatial variability of subsurface soil properties in permafrost topography and argued that permafrost does not conform to Tobler's first law [18]. Lee examined the relationship between distance and destination choice of international leisure traveller activity in Hong Kong over a decade, clarified the relationship between time variation and distance decay [19]. Gao analyzed the Spatial heterogeneity in distance decay of using bike sharing in Shanghai [21]. Hammond examined the fit of logarithmic, negative exponential, and quadratic decay functions to the distribution of the distances travelled to offend by a sample of 70 prolific burglars from the UK, to bring to light the possible psychological and behavioural processes inherent in offending distance decay [23]. However, in the current era of highly developed information technology, does Tobler's first law of geography apply to virtual space? In recent years, empirical efforts attempting to contribute to this debate have utilized emerging data sources derived from location-based services [24,25]. Online social networks [26–31] and mobile communication networks [32–38] are drawing particular attention from researchers. For example, Liu proposed a method for capturing "relatedness between geographical entities" based on the co-occurrences of their names on web pages [26]. Yuan explored the spatial decay effect in mass media and location-based social media [31]. Gao discovered that a high correlation exists between phone users' movements in physical space and phone-call interaction in cyberspace [34]. Kang identified the distance decay effect in intercity mobile communications of China using a subnet data set [35]. But few relevant studies have been conducted based on internet attention. Therefore, this paper attempts to discuss this issue based on the perspective of internet attention.

A review of the literature reveals that existing studies on internet attention tend to be based on Google Trends (frequency of searches and related statistics for a keyword on the Google platform) or the Baidu Index (a data-sharing platform based on Baidu's massive internet user behavior data, based on which keyword search trends can be studied and thus insights can be gained into the changing needs of internet users). Scholars have used Google Trends and the

Baidu Index to research issues such as the correlation between crowding levels and flu outbreaks [39], virus forecasting [40], the relationship between economic growth and ecology [41], the image of geographic locations in virtual environments [42], sales forecasting [43], tourism demand forecasting [44–46], and tourism destination selection behavior [47]. Numerous fields such as medicine, economics, ecology, society, and marketing are involved. Through the existing research, it can be found that the perspective based on internet attention is conducive to observing the flow direction of information in cyberspace and highlighting its spatial differentiation characteristics, so as to summarize and extract the hidden geographic laws.

In summary, this paper attempts to take the Asian elephant northern migration event in southwest China, which has attracted widespread attention at home and abroad, as a case study. It has the characteristics of immediacy, high interest, and a long time span, which can reflect spatial and temporal differences and induce geographic patterns. Using the Baidu index to obtain the internet attention index for the event from April 16 to August 5, 2021, in all provinces (municipalities and autonomous regions). Subsequently, we use spatial visualization and spatial autocorrelation to explore its spatial and temporal evolution characteristics and introduce the geo-detector method to analyze the influencing factors. We tried to discuss the question: Does Tobler's first law of geography apply to internet attention? This study helps to further expand the research outreach of information geography and strengthen the academic community's knowledge of the applicability of Tobler's first law of geography.

## Methods and data

### Methods

**Space autocorrelation.**   Global spatial autocorrelation describes the average degree of association between all spatial units within a study space and its significance, thereby emphasizing the overall spatial relationship. The corresponding formula is [48,49]:

$$Moran's I = \frac{\sum_{i=1}^{n} \sum_{j=1}^{n} w_{ij}(x_i - \bar{x})(x_j - \bar{x})}{S^2 \sum_{i=1}^{n} \sum_{j=1}^{n} w_{ij}} \tag{1}$$

where $n$ indicates the number of provinces (municipalities and autonomous regions) in this study and $w_{ij}$ indicates the spatial weight, $S^2 = \frac{1}{n}\sum_{i=1}^{n}(x_i - \bar{x})^2$, $\bar{x} = \frac{1}{n}\sum_{i=1}^{n} x_i$.

Since the level of spatial autocorrelation often has local differences, the global spatial autocorrelation cannot describe the local spatial differences. Therefore, to better portray the changes and distribution of spatial relationships in space, it is necessary to use the local spatial autocorrelation method for detection; this ensures that the local spatial autocorrelation value centered on the spatial unit is obtained. The corresponding formula is:

$$I_i = Z_i \sum_{i=1}^{n} W_{ij} Z_j \tag{2}$$

where $W_{ij}$ indicates the spatial weight, and $Z_i$ and $Z_j$ are the normalized values of $i$ and $j$ observation space units, respectively.

**Geo-detector.**   Geo-detector is a statistical method used to detect spatial heterogeneity and reveal the influencing factors behind it; it is used for factor detection, interaction detection, risk area detection, and ecological detection [50]. We applied this method to detect the main factors affecting the spatial heterogeneity of the network concerns. The formula used is as follows:

$$P = 1 - \frac{1}{N\alpha^2} \sum_{h-1}^{L} N_h \alpha_h^2 \tag{3}$$

**Table 1. Judgment method of two–factor interaction.**

| Judgment Criteria | Interaction results |
|---|---|
| $q (x_1 \cap x_2) < Min (q (x_1), q (x_2))$ | Non-linear weakening |
| $Min (q (x_1), q (x_2)) < q (x_1 \cap x_2) < Max (q(x_1), q(x_2))$ | Single-factor nonlinear weakening |
| $q (x_1 \cap x_2) > Max (q(x_1), q(x_2))$ | Two-factor enhancement |
| $q (x_1 \cap x_2) = q (x_1)+q(x_2)$ | Independent |
| $q (x_1 \cap x_2) > q (x_1)+q(x_2)$ | Non-linear enhancement |

where $L$ is the classification of the dependent variable $C$ or the independent variable $T$, $N_h$ is the number of partitioned cells, $\alpha_h^2$ is the variance of partition $h$, $N$ is the overall number of cells in the study area, $\alpha^2$ is the variance of the study area as a whole, $P$ is the explanatory strength of the independent variable on the spatial differentiation of the dependent variable, and the explanatory strength of the independent variable on the spatial differentiation of the dependent variable is positively correlated with the size of the P-value whose range is [0, 1].

We also applied the interaction detection function in the geo-detector to identify whether two different influencing factors $X_s$ together enhance or weaken the explanation of the dependent variable compared to a single factor, as shown in Table 1.

## Data

**Study subject.** The Asian elephant northern migration event began on April 16, 2021, when 17 wild Asian elephants from the Mengyangzi Reserve in Xishuangbanna, Yunnan Province left their original habitat and migrated northward through eight counties in the three prefectures of Yuxi, Honghe, and Kunming in Yunnan Province. According to data released by the provincial command of Yunnan's northward Asian elephant herd safety prevention work, to monitor the dynamics of Asian elephant herds and ensure the safety of humans and elephants, the local authorities of Yunnan Province dispatched more than 25,000 police officers and staff, 973 drones, and more than 15,000 emergency vehicles. More than 150,000 people were evacuated, and nearly 180 tons of food were put out for the elephants. The migration disrupted the normal order of day-to-day life and production in the region, causing huge economic losses. This has been the most damaging wildlife incident in recent years in China. Naturally, it has become a hot topic for public attention and opinion at local and global scales.

**Data source.** Internet attention data for the Asian elephant northern migration event were obtained from the Baidu index. First, the search key phrase is formulated by combining three fields: the first field is the "region" field, i.e. the place where the northern migration of Asian elephants occurred, "Yunnan"; the second field is the "subject" field, i.e. the subject of the northern migration of Asian elephants, such as "Asian elephants" and "elephant herd"; the third field is the "behavior" field, i.e. the behavior carried out by the subject, such as "northern migration". Second, the search time was clarified. According to information released by the provincial command of Yunnan's northward Asian elephant herd safety prevention work, the herd originally lived in the Xishuangbanna Mengyangzi Reserve, where they first moved to Mojiang County, Pu'er City, in December 2020, but remained within their traditional habitat. The start of the northward migration of the herd outside of its traditional habitat occurred on April 16, 2021. Therefore, April 16, 2021, was set as the starting date of data collection; the data were collected in periods of "weeks", with the deadline being the week when the data collection was completed. Search phrases were constructed and the search time was specified to achieve accurate collection of internet attention data and avoid noisy data. Ultimately, we collected internet attention data for a total of 16 weeks from April 16 to August 5, 2021.

The data for each influencing factor was derived from the China Statistical Yearbook, China Rural Statistical Yearbook, China Tertiary Industry Statistical Yearbook, 31 provinces (municipalities and autonomous regions), national economic and social development statistical bulletins, Baidu maps, and other online materials.

## Spatial and temporal characteristics

In this part of the study, we try to observe whether the internet attention of a region is stronger when it is closer to the region where the event took place by summarizing the temporal and spatial characteristics.

### Analysis of the time-series characteristics

Based on the internet attention statistics of the elephants' northward migration event, we plotted the temporal evolution of the national and sub-regional events (east, central, and west) to analyze their temporal characteristics. As shown in Fig 1, the time series evolution curves of the whole country (east, central, and west regions) show a "single peak" pattern of rising and then falling, which can be distinguished into three stages: latent, intense, and receding. The latent stage of attention lasted from the first week to the fifth week. At this stage, due to the initial migration of Asian elephants to the north, the event had not yet aroused widespread public concern (apart from the professional departments); the internet attention curve also did not show large fluctuations, showing stable and low characteristics. With the progression of the event, internet attention began to enter the critical stage. Starting from the 6th week, the major mainstream official media (People's Daily, Xinhua, CCTV News, etc.), short videos (Tik Tok, Kwai, etc.), and portals (Weibo, Baidu, Tencent, etc.) issued many reports and propaganda on the event, which quickly aroused extensive discussions amongst the populace with the internet attention continuing to escalate the trend that continued until the 8th week (when it reached its peak). After the 9th week, the fervor of the event gradually subsided across all media, frequency of publicity reports decreased, and the internet attention curve showed a downward trend, thus entering the stage of receding attention. Comparing the fervent and receding stages, we found that the receding curve was smoother and lasted longer than the upward curve, indicating that the development of online public opinion quickly climbs to a peak over a short period of time, while the receding speed of public opinion is slower and lasts longer. In addition, it can also be discovered that the values of internet attention at all time points of the fervent and recession stages are greater than those of the latent stage, indicating that the publicity and coverage of events by mainstream social media have a value-added diffusion effect on internet attention, which is most evident during the intensive period of media coverage and continues to exist during the recession stage when media attention shifts.

In addition, the Asian elephant northward migration event occurred in the western region. However, the highest internet attention was in the east, followed by that in the west, and lowest in the center, which does not conform to Tobler's first law of geography. This may be related to the higher level of economic development and more developed information network in the eastern region. We can also speculate that in the era of information, the rapid spread of the internet will largely overcome the obstacle of geographical and spatial distance for information access and dissemination.

### Analysis of the spatial evolution characteristics

**Spatial visualization.** To present the regional differences in a more detailed way, the natural breaks (Jenks) method in the ArcGIS software was used to classify the internet attention of 31 provinces (municipalities and autonomous regions) into five levels: highest, higher,

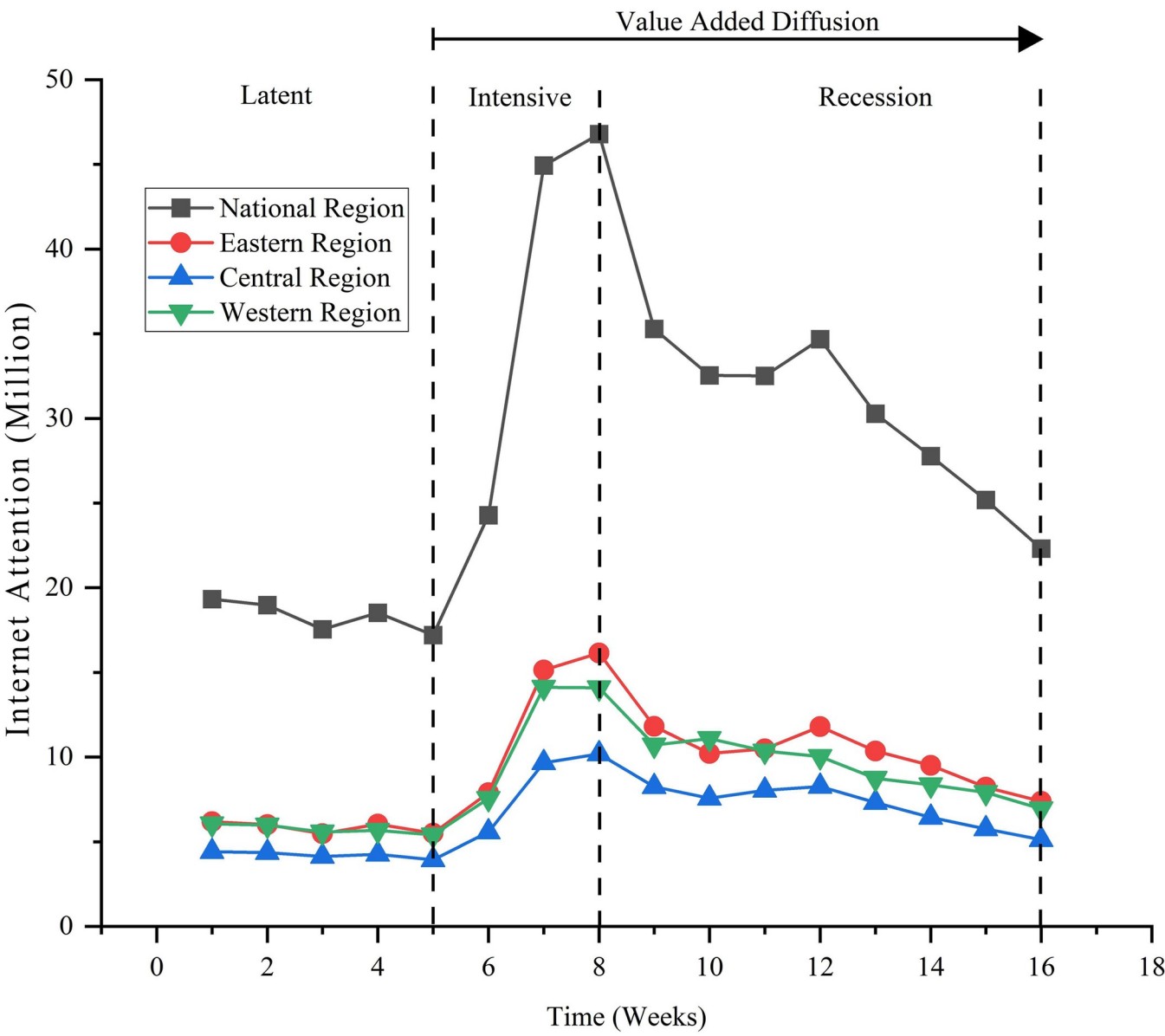

**Fig 1. Time series evolution of internet attention nationwide and by region (East, Central and West China).**

medium, lower, and lowest. Spatial visualization was used to further analyze the variation characteristics of the internet attention at the provincial level. As shown in Fig 2, the regions with the highest internet attention were only two provinces, Yunnan and Guangdong; the regions with higher internet attention cover Sichuan, Beijing, Zhejiang, Shanghai, Jiangsu, Shandong, Henan, and other places along the east coast. The areas with intermediate internet attention were mainly located in Hubei, Hunan, Jiangxi, Anhui, and other central regions, and the areas with lower and lowest internet attention were concentrated in the northeast (Heilongjiang, Jilin, Inner Mongolia, etc.) and northwest regions (Xinjiang, Tibet, Qinghai, Gansu, Ningxia, etc.) of China. Although there were individual regions (e.g., Guangxi, Jiangxi, Ningxia, and Inner Mongolia) where the internet attention increased during the study period, the overall pattern of spatial divergence was relatively stable. We can find that Yunnan has the highest

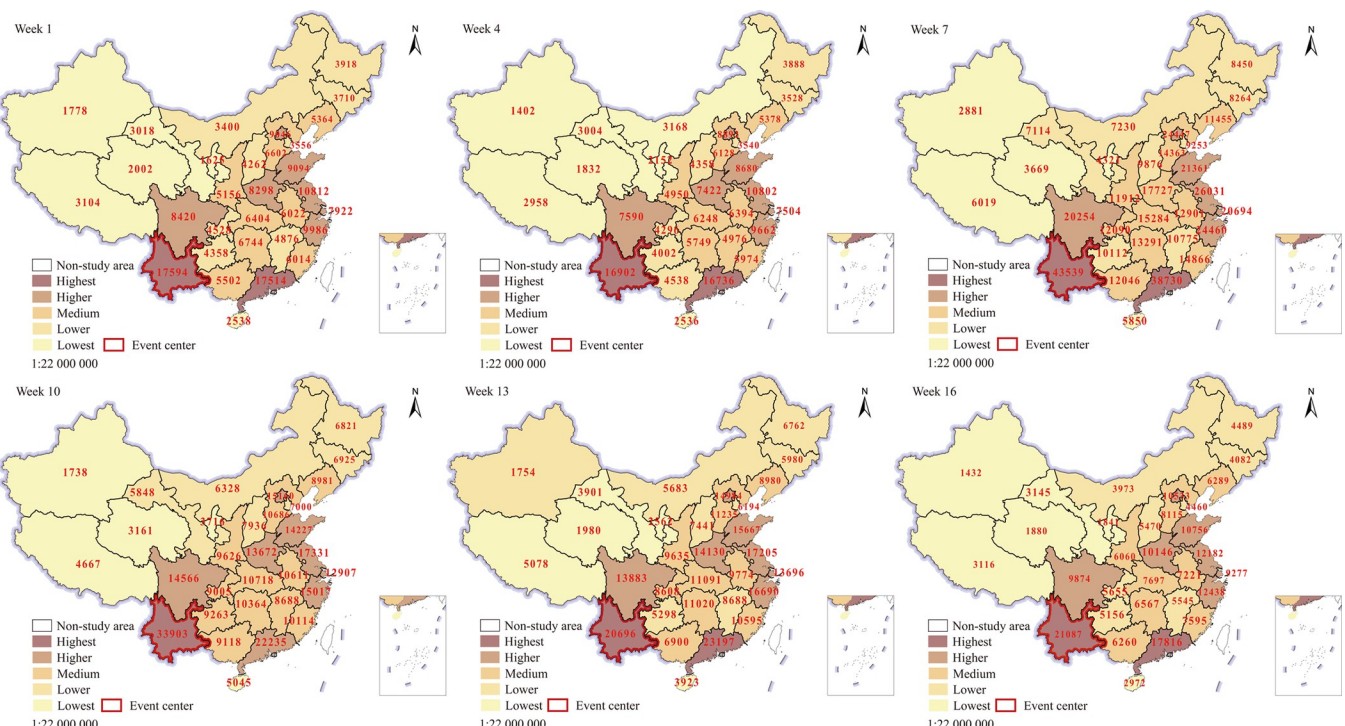

**Fig 2. Spatial representation of internet attention in 31 provinces (municipalities directly under the Central Government and autonomous regions).** The underlying layer is quoted from the Sky Map website (https://www.tianditu.gov.cn/), the standard map review number is GS (2019) 1833.

internet attention as the region where the event took place. However, Guangdong, Jiangsu, Zhejiang, Shandong, and Henan provinces, which are far away from Yunnan, have higher internet attention than Sichuan, Guangxi, and Guizhou, which are adjacent to Yunnan. This situation is not by Tobler's first law of geography. It suggests that the spatial divergence of internet attention is not decisively influenced by geographic proximity, and is probably influenced by other factors as well.

Yunnan was always at the top of the list, probably because it is the location of the migration. Being a direct stakeholder, the population of Yunnan was extremely interested in this event because of their "local" identity. Guangdong ranked with Yunnan as the region with the highest internet attention, partly because of its relatively large population (126 million people) and developed economy and society. Sichuan Province, which is adjacent to Yunnan Province that is rich in biodiversity, is the main habitat and reserve of China's national treasure: giant pandas; thus, it emphasizes on the protection of giant pandas and other wild animals. Therefore, people in the region are more concerned and aware of the need for protecting wild animals. The levels of economic development, urbanization, internet penetration, and population in Beijing, Shanghai, and Jiangsu are generally high, providing a good foundation for people to browse on the internet and thus have a high level of internet attention. Regions with relatively low internet attention ratings either have a long distance between them and Yunnan Province or have characteristics such as low population, average economic development level, low urbanization level, low network penetration rate, and low education level of residents. It is easy to understand that geographic proximity still has some influence on the spatial divergence of internet attention, but it is not in a dominant factor and is also subject to the combined effect of other factors.

**Table 2. Moran's I value for internet attention.**

|  | Week 1 | Week 4 | Week 7 | Week 10 | Week 13 | Week 16 |
|---|---|---|---|---|---|---|
| **Moran's I** | 0.0834 | 0.0829 | 0.1007 | 0.1262 | 0.1596 | 0.1234 |
| **z-score** | 1.5542 | 1.5444 | 1.7556 | 2.1931 | 2.3701 | 2.0335 |
| **P-value** | 0.0580 | 0.0610 | 0.0450 | 0.0220 | 0.0170 | 0.0310 |

*Note*. Moran's I index takes values between [−1,1], the closer the value is to −1, the stronger the negative correlation; the closer it is to 1, the stronger the positive correlation; and close to 0, it indicates that the spatial agglomeration phenomenon is not significant.

**Spatial clustering.** GeoDa software was applied to calculate the global Moran's I index of internet attention for the migration event (Table 2). The results showed that the Moran's I index at the time points examined showed values in the interval [0.0834, 0.1596], which were positive; however, the values were not high and tended to increase over time, passing the 5% and 10% significance tests, respectively. This reveals that internet attention within the study area exhibits a statistically significant positive spatial correlation; that is, there is a certain spatial clustering characteristic, although the positive correlation is relatively insignificant.

To further explore the local spatial association characteristics, a Lisa agglomeration map was measured and drawn using GeoDa software (Fig 3). In terms of clustering patterns, three main types of clustering patterns existed during the study period, namely: high-high clustering (high internet attention areas are adjacent to each other with small spatial differences in attribute characteristics), low-high clustering (low internet attention areas are adjacent to high internet attention areas with large spatial differences in attribute characteristics), and low-low clustering (low internet attention areas are adjacent to each other with small spatial differences in attribute characteristics). The high-high aggregation area shows a trend of "diffusion-

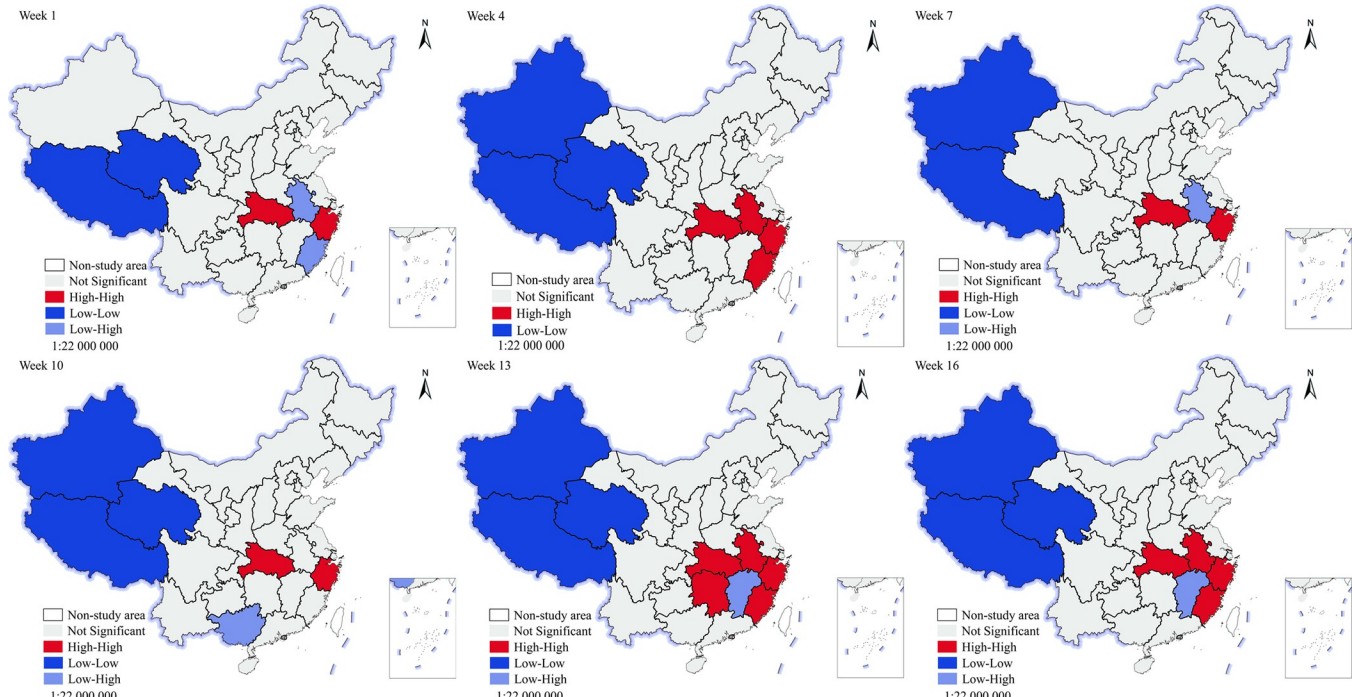

**Fig 3. Lisa's clustering chart of internet attention.** The underlying layer is quoted from the Sky Map website (https://www.tianditu.gov.cn/), the standard map review number is GS (2019) 1833.

shrinking-rediffusion" during weeks 1–10. There are only two regions in Hubei and Zhejiang in week 1, and two regions in Anhui and Fujian were added in week four; however, these two regions dropped out of a high-high aggregation area in week seven. Although the spatial distribution of the high-high aggregation area increased in the first and middle periods, relatively stable high-high aggregation areas were formed mainly around Hubei and Zhejiang provinces. During weeks 11–16, the high-high aggregation area spread again and formed agglomerative clusters, with Hunan, Hubei, Anhui, Zhejiang, and Fujian as the main members. The distribution of low-high aggregation areas differed significantly in weeks 1–10, jumping continuously in regions such as Fujian, Anhui, and Guangxi until a stable low-high aggregation area centered on Jiangxi was formed after week 13. The spatial distribution of low-low aggregation areas did not change significantly; only significant transitions in individual regions (Xinjiang, Qinghai) occurred during weeks 1–7. However, from the 10th week onwards, clusters were formed in low-lying areas including Tibet, Xinjiang, and Qinghai.

We can observe more clearly that the high-high clustering areas are mainly concentrated on the southeast coast of China, far from Yunnan, where the event took place. This again shows that Tobler's first law of geography does not apply to internet attention.

## Factors influencing the spatio-temporal evolution

In the spatial analysis above, we initially inferred that Tobler's first law of geography does not apply to internet attention and that the spatial divergence of internet attention is likely to be influenced by a combination of many factors. Therefore, in this part of the study, we introduce the method of Geo-detector to analyze the extent to which each factor, including geographic proximity, affects the spatial divergence of internet attention when many factors are combined.

### Impact factor selection

Combining the characteristics of the Asian elephant northward migration event, drawing on relevant research results, and considering the compatibility, comprehensiveness, scientific significance, and data accessibility of the indicators, an index system affecting the spatial and temporal heterogeneity of the attention of the Asian elephant northward migration event network was constructed (Table 3). The index system is explained in terms of five aspects: geographical

**Table 3. Index system of influencing factors of internet attention.**

| | Number | Detection factor | Representative Indicator | Property |
|---|---|---|---|---|
| **Geographical location** | $X_1$ | Geographic proximity | Shortest land transportation distance to Yunnan Province (km) | - |
| **Regional development** | $X_2$ | Economic development level | GDP per capita (yuan) | + |
| | $X_3$ | Industrial development level | Gross domestic product of primary, secondary and tertiary industries (billion yuan) | + |
| | $X_4$ | Urbanization level | Urban population to total population ratio (%) | + |
| **Ecological environment** | $X_5$ | Forest coverage | Forest cover area to total area ratio (%) | + |
| | $X_6$ | Farmland coverage | Agricultural land area to total area ratio (%) | + |
| | $X_7$ | The number of nature reserves | Number of nature reserves (pcs) | + |
| **Demographic characteristics** | $X_8$ | Population | Total regional population (million people) | + |
| | $X_9$ | Population age structure | Ratio of population aged 15–64 to total population (%) | + |
| | $X_{10}$ | Educational attainment | Ratio of population with education level above high school compared to total population (%) | + |
| **Network and publicity** | $X_{11}$ | The degree of information technology | Internet penetration rate (%) | + |
| | $X_{12}$ | The strength of news media publicity | Baidu information concern index (-) | + |

location, regional development, ecological environment, demographic characteristics, network and publicity.

① Geographic location is represented by geographic proximity ($X_1$), and, ceteris paribus, areas closer to the center of the event (Yunnan Province) are likely to be of higher relative concern.

② Regional development covers three major detection factors: economic development level ($X_2$), industrial development level ($X_3$), and urbanization level ($X_4$). The level of economic development ($X_2$) reflects, to a certain extent, the quality of life and consumption ability of the people in the study area, and the internet penetration rate tends to be positively related to the level of economic development. The industrial development level ($X_3$) mainly refers to the development of industries such as agriculture, industry, and tourism, which may encroach on and destroy wildlife habitats, thus increasing the probability of animal incidents in the region, and the local population may pay more attention to such incidents. The level of urbanization ($X_4$) is the degree of gradual transformation of society from a traditional rural type to a modern urban type. Generally, areas with higher levels of urbanization will have better infrastructure and may have higher levels of local network construction.

③ The ecological environment consists of three major detection factors: forest coverage ($X_5$), farmland coverage ($X_6$), and the number of nature reserves ($X_7$). In areas with high forest coverage ($X_5$) and the greater the spatial overlap between the main areas inhabited by wild animals and areas where humans live and work, the more attention will be paid to such incidents. Intensive farming areas ($X_6$) generally have a higher probability of animal damage of food crops, and therefore the local population may be more concerned about animal disturbance. In areas with many nature reserves ($X_7$), the public may be more aware of wildlife protection and more concerned about the occurrence of wildlife-related incidents.

④ Demographic characteristics mainly involve three detection factors: population ($X_8$), population age structure ($X_9$), and educational attainment ($X_{10}$). A high population ($X_8$) facilitates the dissemination of information and reinforces the active online search behavior of the public. The age structure of the population ($X_9$) reflects the concentration of internet users in China, especially in the age range of 15–64 years. Educational attainment ($X_{10}$) indicates the merit of the population of a region. Those with higher educational attainment have a stronger desire to learn about new things and topical events and are more capable of understanding and sharing current events on the internet.

⑤ Network and publicity are mainly related to the degree of information technology ($X_{11}$) and the strength of news media publicity ($X_{12}$). Areas with a high degree of information technology ($X_{11}$) are also more rapid at searching and receiving the information dynamics of events. The strength of news media publicity ($X_{12}$) reflects the attention and coverage of relevant credentials, which is shown in this study through the Baidu Information Attention Index.

## Factor detection analysis

Using the Natural Breaks (Jenks) method in ArcGIS software to convert the influencing factors from continuous variables to discrete variables, and applying Geo-detector to detect the degree of influence of each influencing factor on the internet attention from two dimensions of temporal evolution and spatial differentiation, and referring to related studies, the factors that passed the 1% significance test were considered core influencing factors, and those that passed

**Table 4. Detection results of internet attention time series evolution factor.**

| Factor | Week 1 | | Week 4 | | Week 7 | | Week 10 | | Week 13 | | Week 16 | |
|---|---|---|---|---|---|---|---|---|---|---|---|---|
| | *q* | *p* | *q* | *p* | *q* | *p* | *q* | *p* | *q* | *p* | *q* | *p* |
| $X_1$ | 0.333 | 0.771 | 0.337 | 0.764 | 0.368 | 0.709 | 0.523 | 0.475 | 0.182 | 0.909 | 0.375 | 0.720 |
| $X_2$ | 0.204 | 0.561 | 0.221 | 0.522 | 0.241 | 0.492 | 0.129 | 0.784 | 0.285 | 0.440 | 0.202 | 0.573 |
| $X_3$ | 0.550* | 0.062 | 0.551* | 0.063 | 0.504* | 0.094 | 0.392 | 0.241 | 0.720*** | 0.000 | 0.524* | 0.073 |
| $X_4$ | 0.161 | 0.493 | 0.171 | 0.465 | 0.177 | 0.467 | 0.107 | 0.700 | 0.216 | 0.356 | 0.152 | 0.526 |
| $X_5$ | 0.206 | 0.349 | 0.204 | 0.355 | 0.204 | 0.345 | 0.229 | 0.260 | 0.251 | 0.229 | 0.226 | 0.277 |
| $X_6$ | 0.124 | 0.500 | 0.117 | 0.532 | 0.114 | 0.545 | 0.148 | 0.413 | 0.131 | 0.464 | 0.135 | 0.455 |
| $X_7$ | 0.212 | 0.329 | 0.218 | 0.322 | 0.212 | 0.340 | 0.193 | 0.395 | 0.207 | 0.396 | 0.210 | 0.336 |
| $X_8$ | 0.429* | 0.072 | 0.406* | 0.090 | 0.353 | 0.141 | 0.335 | 0.150 | 0.553** | 0.013 | 0.423* | 0.071 |
| $X_9$ | 0.081 | 0.757 | 0.091 | 0.717 | 0.083 | 0.745 | 0.086 | 0.736 | 0.085 | 0.746 | 0.076 | 0.776 |
| $X_{10}$ | 0.067 | 0.855 | 0.089 | 0.781 | 0.086 | 0.805 | 0.091 | 0.778 | 0.118 | 0.720 | 0.092 | 0.775 |
| $X_{11}$ | 0.162 | 0.560 | 0.166 | 0.555 | 0.169 | 0.527 | 0.136 | 0.602 | 0.134 | 0.658 | 0.136 | 0.619 |
| $X_{12}$ | 0.768*** | 0.000 | 0.761*** | 0.000 | 0.834*** | 0.000 | 0.886*** | 0.000 | 0.827*** | 0.000 | 0.829*** | 0.000 |

Note: *q* is the influence value, *p* is the significance value, and *, **, and *** indicate passing the significance test of 10%, 5%, and 1%, respectively.

the 5% and 10% significance tests were considered important influencing factors, and the detection results are shown in Tables 4 and 5.

## Temporal evolution

According to the detection results in Table 4, among the 12 detection factors, only industrial development level ($X_3$), population ($X_8$) and the strength of news media publicity ($X_{12}$) passed the significance test, indicating that the above three factors are the most important factors influencing the temporal evolution of internet attention. Geographical proximity ($X_1$) did not pass the significance test, indicating that it did not play a significant role.

In order of influence and importance: the strength of media publicity ($X_{12}$) > industrial development level ($X_3$) > population ($X_8$). Among them, the strength of news media publicity ($X_{12}$) passed the 1% significance test in all examined nodes, and the q-value was stronger than

**Table 5. Sub–regional internet attention factor detection results.**

| Factor | East | | Central | | West | |
|---|---|---|---|---|---|---|
| | *q* | *p* | *q* | *p* | *q* | *p* |
| $X_1$ | 0.107 | 0.944 | 0.769 | 0.482 | 0.915*** | 0.023 |
| $X_2$ | 0.322 | 0.828 | 0.653 | 0.674 | 0.191 | 0.819 |
| $X_3$ | 0.857* | 0.058 | 0.987** | 0.020 | 0.503 | 0.472 |
| $X_4$ | 0.694 | 0.528 | 0.553 | 0.944 | 0.079 | 0.957 |
| $X_5$ | 0.262 | 0.839 | 0.540 | 0.873 | 0.472 | 0.424 |
| $X_6$ | 0.326 | 0.723 | 0.571 | 0.597 | 0.436 | 0.463 |
| $X_7$ | 0.272 | 0.876 | 0.604 | 0.871 | 0.326 | 0.636 |
| $X_8$ | 0.754 | 0.155 | 0.960** | 0.036 | 0.464 | 0.405 |
| $X_9$ | 0.499 | 0.836 | 0.749 | 0.496 | 0.130 | 0.910 |
| $X_{10}$ | 0.156 | 0.935 | 0.510 | 0.918 | 0.319 | 0.767 |
| $X_{11}$ | 0.260 | 0.856 | 0.717 | 0.817 | 0.420 | 0.623 |
| $X_{12}$ | 0.934** | 0.027 | 0.992*** | 0.000 | 0.964*** | 0.000 |

Note: *q* is the influence value, *p* is the significance value, and *, **, and *** indicate passing the significance test of 10%, 5%, and 1%, respectively.

the other two factors, indicating that the strength of media publicity ($X_{12}$) was the core influencing factor. The public has initial contact and understanding of the northern migration of Asian elephant events based on media propaganda reports, and gradually develops interest and curiosity with the increasing frequency of reports and continuous disclosure of details. Finally, interest and curiosity eventually evolve into active online search behavior. The industrial development level ($X_3$) passed the significance tests of 1% at week 13, and 10% at most of the remaining time points; thus, it can be classified as an important influencing factor. The industrial development level ($X_3$) passed the significance test, indicating that the public's concern about wildlife incidents and conservation increased with the level of industrial development. Population ($X_8$) passed the 5% significance test in week 13, and only passed the 10% significance test at most time points, so it can only be classified as a important influencing factor, which also verifies the conclusion that Guangdong is ranked as a high internet concern region above because it has a high population size.

## Spatial differentiation

We divided the study area into three regions: East, Central, and West, and tested the influence factors separately to further clarify the influence of geographical proximity on the spatial divergence of internet attention. Yunnan, where the event took place, is located in the western region. Therefore, the western region is the closest to the event site, the central region is the next closest, and the eastern region is the farthest. According to the detection results in Table 5, geographic proximity ($X_1$) passed the significance test in the western region but did not gain support in the eastern and central regions. However, the results of "3.1 Analysis of the Time-Series Characteristics" above show that the eastern region had higher internet attention than the western region, and the western region had higher internet attention than the central region. Combining the two results, it is thought that geographic distance has a positive effect on internet attention, but this effect is often limited to a certain range. When a certain threshold is exceeded, the degree of influence of geographic distance on internet attention may be weakened by the intervention of other influencing factors.

In addition, the strength of news media publicity ($X_{12}$) and industry development level ($X_3$) in the eastern and central regions passed the significance test. In the west, the strength of media publicity ($X_{12}$) also passed the significance test. The strength of media publicity ($X_{12}$) topped the explanatory ratings of many influencing factors in all three regions, indicating that regional differences did not weaken the effect of its on internet attention, further validating the status of the strength of media publicity ($X_{12}$) as a core influencing factor. The level of industrial development in the eastern and central regions of China is more developed than in the western regions; therefore, the industrial development level ($X_3$) is considerably better at explaining internet attention in the eastern and central regions.

## Factor interaction

As mentioned above, the spatial variation of internet attention is influenced by the combined effect of multiple factors, so is the degree of influence of multi-factor interaction on the spatial variation of internet attention stronger than that of a single factor? Which factors had the strongest influence when interacting with each other? To answer these questions, this study used the factor interaction detection module in the geo-detector to conduct two-factor interaction detection. As shown in Tables 6 and 7, the influence of both two-factor interactions was greater than that of the single factor, with 61% of the interaction types being nonlinearly enhanced and 39% being two-factor enhanced. Of these, population age structure ($X_9$) had the largest effect on the interaction; among the 11 factors, only one factor, the strength of news

**Table 6. Influence of factors interaction.**

|        | $X_1$ | $X_2$ | $X_3$ | $X_4$ | $X_5$ | $X_6$ | $X_7$ | $X_8$ | $X_9$ | $X_{10}$ | $X_{11}$ | $X_{12}$ |
|--------|-------|-------|-------|-------|-------|-------|-------|-------|-------|----------|----------|----------|
| $X_1$    | 0.354 |       |       |       |       |       |       |       |       |          |          |          |
| $X_2$    | 0.835 | 0.332 |       |       |       |       |       |       |       |          |          |          |
| $X_3$    | 0.985 | 0.803 | 0.663 |       |       |       |       |       |       |          |          |          |
| $X_4$    | 0.857 | 0.548 | 0.958 | 0.177 |       |       |       |       |       |          |          |          |
| $X_5$    | 0.604 | 0.704 | 0.761 | 0.761 | 0.251 |       |       |       |       |          |          |          |
| $X_6$    | 0.598 | 0.673 | 0.745 | 0.600 | 0.451 | 0.169 |       |       |       |          |          |          |
| $X_7$    | 0.791 | 0.550 | 0.792 | 0.419 | 0.746 | 0.518 | 0.134 |       |       |          |          |          |
| $X_8$    | 0.932 | 0.830 | 0.773 | 0.990 | 0.698 | 0.718 | 0.725 | 0.510 |       |          |          |          |
| $X_9$    | 0.727 | 0.601 | 0.986 | 0.644 | 0.913 | 0.510 | 0.579 | 0.791 | 0.080 |          |          |          |
| $X_{10}$   | 0.629 | 0.752 | 0.824 | 0.408 | 0.419 | 0.539 | 0.415 | 0.737 | 0.474 | 0.131    |          |          |
| $X_{11}$   | 0.517 | 0.882 | 0.832 | 0.854 | 0.630 | 0.690 | 0.799 | 0.924 | 0.777 | 0.673    | 0.129    |          |
| $X_{12}$   | 0.990 | 0.922 | 0.960 | 0.942 | 0.976 | 0.946 | 0.963 | 0.982 | 0.923 | 0.912    | 0.938    | 0.867    |

Note: Yellow in the table represents two–factor enhancement and blue represents non–linear enhancement.

media publicity ($X_{12}$), had a two-factor enhancement type of interaction with population age structure, whereas the others were all non-linearly enhanced. The type of interaction between the strength of news media publicity ($X_{12}$) and the other 11 factors is a two-factor enhancement, and they do not promote each other much. The reason for this may be that the single factor of the strength of news media publicity ($X_{12}$) is too influential, resulting in a "shielding effect" and "ceiling effect". Specifically, the interaction influence of the strength of news media publicity ($X_{12}$) and geographic proximity ($X_1$) was 0.990, ranking first among all factors of interaction influence, indicating that the two have the strongest driving effect on the spatial divergence of internet attention when they interact together. In addition, although the strength of news media publicity ($X_{12}$) does not contribute significantly to other factors when interacting with them, the interaction effect is significant (all above 0.900), further verifying that the strength of news media publicity ($X_{12}$) is the core influencing factor affecting the spatial variation of internet attention.

## Discussion and conclusion

### Discussion

Tobler's first law of geography states that everything is related to everything else, but near things are more related to each other. It emphasizes the key role played by distance in the connection between things. Therefore, in the material world, this law has been more widely applied because there is a real spatial distance between things in the material world. However, in the current internet era, the concept of distance seems to gradually blur or even disappear in the internet space. So, does Tobler's first law of geography still apply to things in the internet space? This question remains to be continuously explored and discussed. Therefore, this study attempts to discuss the applicability of Tobler's first law of geography to internet attention in internet space, using the Asian elephant northern migration event as a research case. It helps

**Table 7. Statistics of the types of interactions between the factors and other factors.**

|                         | $X_1$ | $X_2$ | $X_3$ | $X_4$ | $X_5$ | $X_6$ | $X_7$ | $X_8$ | $X_9$ | $X_{10}$ | $X_{11}$ | $X_{12}$ |
|-------------------------|-------|-------|-------|-------|-------|-------|-------|-------|-------|----------|----------|----------|
| Two-factor enhancement  | 4     | 4     | 8     | 2     | 6     | 4     | 2     | 5     | 1     | 3        | 2        | 11       |
| Non-linear enhancement  | 7     | 7     | 3     | 9     | 5     | 7     | 9     | 6     | 10    | 8        | 9        | 0        |

to answer the pessimistic arguments about the "death of distance" and "the end of geography" in the internet era and further deepens the understanding of Tobler's first law of geography.

We found that the rapid spread of information technology has indeed broken the distance barrier in the process of people's active access to information to a certain extent, distance does not have a decisive influence on internet attention, and Tobler's first law of geography does not fully apply to internet attention. Of course, this does not mean that spatial distance does not have any influence on internet attention. On the internet, regardless of the distance from the information source, the time and efficiency associated with information access are often similar. However, under the condition of relative geographical proximity, influenced by "identity" and "interest-related", the phenomenon of "distance decay" is still characterized; that is, "the closer the distance → more relevant interest → higher internet attention". Once this relative distance is exceeded and there is no direct correlation of interest other factors, such as news media coverage, population, and internet penetration, which may have a greater impact on internet attention than geographic distance.

In addition, the results of our study can also help provide a reference for internet public opinion management, because internet attention is an indicator that characterizes the active information acquisition behavior of internet users, which is a manifestation of internet public opinion. First, the evolution of internet attention shows the stages of "latent → hot spot → decline". The relevant departments need to achieve a real-time and precise understanding of the development stage of internet public opinion, whether they carry out emergency control of internet public opinion or use the hotness of internet public opinion to conduct relevant publicity activities. However, in the actual practice of internet governance, it is often a passive post-facto action rather than a proactive prevention, which is ultimately due to the lack of knowledge of the regularity of internet public opinion, making it difficult to carry out quantitative predictions. Therefore, there is an urgent need to strengthen the ability to monitor and judge internet public opinions. Mainstream media reports on the event have a value-added diffusion effect on internet attention; this effect is most obvious during the intensive period of mainstream media reports, even if the effect is reduced with the shift of mainstream media attention. Mainstream media reports are a core factor in influencing internet attention; when it interact with geographical proximity, its influence is even more powerful. Thus, in terms of internet public opinion monitoring, it is necessary to pay full attention to combining the use of emerging technologies such as artificial intelligence and big data analysis and deeply integrating internet public opinion collection, data storage, and data mining systems to achieve dynamic monitoring. In terms of internet public opinion research and judgment, a special internet opinion evaluation team with keen insight and judgment should be cultivated and built to realize the scientific, standardized, and professional evaluation and prediction of internet public opinion. To achieve accurate research and judgment of internet public opinion, we must improve the ability to find opportunities from crises and strengthen the scientific application of internet public opinion. For example, in this incident of the elephant migration, Yunnan Province made full use of the internet public opinion and timely launched the slogans of "Elephants yearning for Kunming", "A place where elephants yearn for" and so on, to achieve a worldwide city branding and image building. This has also been a typical example of China's biodiversity conservation achievements during the 15th Conference of the Parties to the Convention on Biological Diversity (CBD), which is a successful experience worth learning from. Naturally, the use of internet public opinion should also take into consideration spatial variability. During this incident, internet attention formed two high-attention areas, namely "southeast" and "southwest". For these two regions, the internet fervor and follow-up effect of this incident can be used to educate people about wildlife conservation and stimulate their awareness of the importance of biodiversity conservation.

Of course, there are some limitations of our study. First, this study analyzes the applicability of Tobler's first law of geography to internet attention, but the internet space is a complex virtual space that also includes other things beyond internet attention, and the study of these things needs to be further explored in the future. Second, this study is a case study, and more cases need to be added to verify it in the future. Finally, this study is an indirect discussion of the applicability of Tobler's first law of geography to internet attention based on the analysis of the spatial and temporal characteristics and influencing factors of internet attention. Future research can try to use quantitative models to establish more direct arguments for geospatial relationships.

## Conclusion

This study explores the applicability of Tobler's first law of geography to internet attention using the Asian elephant northern migration event as a research case. We found that Tobler's first law of geography does not apply to internet attention, and the distance decay effect is not fully reflected in internet attention. Distance has an effect on internet attention, but this effect is often limited to a certain distance, and when a certain distance threshold is exceeded, the degree of influence of geographic proximity on internet attention is probably no longer significant due to the intervention of other influencing factors. Therefore, the influence of distance on internet attention is not decisive. The factor that really has a decisive influence on internet attention is the strength of news media publicity, and it has the greatest influence on internet attention when the strength of news media publicity interacts with geographic proximity.

## Supporting information

**S1 Table. Internet attention of provinces during week 1 to the week 16.**
(XLSX)

**S2 Table. Factors.**
(XLSX)

## Acknowledgments

The authors acknowledge the two reviewers for their valuable comments on the paper, which have helped greatly improve the paper's quality.

## Author Contributions

**Conceptualization:** Boming Zheng.

**Data curation:** Boming Zheng, Xijie Lin.

**Formal analysis:** Boming Zheng, Xijie Lin.

**Funding acquisition:** Duo Yin, Xinhua Qi.

**Methodology:** Xijie Lin.

**Project administration:** Duo Yin, Xinhua Qi.

**Supervision:** Duo Yin, Xinhua Qi.

**Writing – original draft:** Boming Zheng, Xinhua Qi.

**Writing – review & editing:** Boming Zheng, Xinhua Qi.

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
