## [Decision Letter · Decision Letter 0]

14 Nov 2022

PONE-D-22-25265Does Tobler's first law of geography apply to cyberspace? An exploration based on network attentionPLOS ONE

Dear Dr. XINHUA,

Thank you for submitting your manuscript to PLOS ONE. After careful consideration, we feel that it has merit but does not fully meet PLOS ONE’s publication criteria as it currently stands. Therefore, we invite you to submit a revised version of the manuscript that addresses the points raised during the review process.

We look forward to receiving your revised manuscript.

Kind regards,

Wei Tu

Academic Editor

PLOS ONE

Journal Requirements:

"This research was funded by the National Social Science Foundation of China, grant number 18BJL126"

5. We note that Figures 2 and 3 in your submission contain map images which may be copyrighted. All PLOS content is published under the Creative Commons Attribution License (CC BY 4.0), which means that the manuscript, images, and Supporting Information files will be freely available online, and any third party is permitted to access, download, copy, distribute, and use these materials in any way, even commercially, with proper attribution. For these reasons, we cannot publish previously copyrighted maps or satellite images created using proprietary data, such as Google software (Google Maps, Street View, and Earth). For more information, see our copyright guidelines: http://journals.plos.org/plosone/s/licenses-and-copyright.

a. You may seek permission from the original copyright holder of Figures 2 and 3 to publish the content specifically under the CC BY 4.0 license.  

Reviewers' comments:

Reviewer's Responses to Questions

**Comments to the Author**

1. Is the manuscript technically sound, and do the data support the conclusions?

Reviewer #1: Partly

Reviewer #2: Yes

2. Has the statistical analysis been performed appropriately and rigorously? 

Reviewer #1: Yes

Reviewer #2: Yes

3. Have the authors made all data underlying the findings in their manuscript fully available?

Reviewer #1: Yes

Reviewer #2: No

4. Is the manuscript presented in an intelligible fashion and written in standard English?

Reviewer #1: Yes

Reviewer #2: Yes

5. Review Comments to the Author

Reviewer #1: This study explored an interesting question that does the Tobler's first law of geography apply to cyberspace by using a good example, i.e., the event of elephant migration in southwest China. The manuscript is clearly structured and well written in English. The presented data, methods, and analysis sound reasonable. I like the topic of this study and it sheds a light on the relation of “the Tobler’s first law of geography” and “network attention in cyberspace”.

However, the main issue of this study is that the conclusions are drawn only from analysis toward the event of elephant migration in southwest China. Even though it is a good case for study this topic, the conclusions are still event specific. In other word, one would question if the conclusions drawn in this study are general and solid. Some suggestions are as follows.

(1) The authors are suggested to add “a case study of *** event” in the title. And the event should also be mentioned in the abstract.

(2) Any analysis or conclusion should be carefully expressed throughout the manuscript.

(3) This limitation should also be emphasized in the Discussion section.

(4) Some studies have investigated the distance-decay effect in cyberspace, which are suggested to be mentioned in the manuscript. And therefore, this manuscript should emphasize or narrow down to “network attention” instead of “cyberspace”.

* Liu, Y., Wang, F., Kang, C., Gao, Y. and Lu, Y., 2014. Analyzing Relatedness by Toponym Co-Occurrences on Web Pages. Transactions in GIS, 18(1), pp.89-107.

* Krings, G., Calabrese, F., Ratti, C., & Blondel, V. (2009). A gravity model for inter-city telephone communication networks. Journal of Statistical Mechanics: Theory and Experiment, 7003.

* Gao, S., Liu, Y., Wang, Y., & Ma, X. (2013). Discovering spatial interaction communities from mobile phone date. Transactions in GIS, 17(3), 463-481.

* ...more related literature that discuss distance-decay effect in cyberspace

(5) The authors are suggested to explain the term “network attention” earlier in the manuscript. In addition, “Internet attention” sounds better than “network attention” as “network” has a very broad meaning.

(6) In the Discussion section, the discussion about “whether the traditional laws of geography are applicable to network space-time” should be put in the first place. The limitation of the conclusions should be emphasized (i.e., conclusions are drawn from the case study).

(7) A small mistake in Page 11: “In the west, the strength of media publicity (X11) and geographical proximity (X11) passed the significance test.” The geographical proximity is X12 instead of X11.

Reviewer #2: This study aims to analyze the geographical characterize of media attention of certain event, specifically, focuses on whether the network attention following some geography law “The firs law of geography”, and further investigates the influence of geographical factors on attention. I have some comments as follow:

1. The title of this manuscript is “Does Tobler's first law of geography apply to cyberspace? An exploration based on network attention”, However, the content is little related with the topic. There is no specific experiment to verify the first law of geography, authors only conduct some conclusions from several figures. Therefore, the content does not follow the title.

2. The term ”network attention ” is easy to have ambiguity. The network usually refers to complex network. In fact, this study just refers to news attention.

3. From Figure 2, the six maps are same, it can not see the difference among these figures, which makes these maps no significant. Usually, I’m more concerned about the changing value of attention over time. For example, the attention in Province A and B are 100 and 1000 respectively on week 4, then the attention on week 7 are 400 and 1100. We can see that the changing value is 300 for Province A, while the changing value is only 100 for Province B. if we only visualize the total attention, it may miss the process of change.

4. In impact factor, the population should play an important role on attention, the more population is, the more attention should be. The population density could not reflect this phenomenon.

5. A conclusion should be given in the manuscript.

6. PLOS authors have the option to publish the peer review history of their article (what does this mean?). If published, this will include your full peer review and any attached files.

Reviewer #1: No

Reviewer #2: No

---

## [Author Response · Author response to Decision Letter 0]

10 Jan 2023

We are very sincerely grateful to the two reviewers for their valuable comments, which have greatly helped improve the paper's quality. The authors have carefully and conscientiously revised the paper based on the reviewers' comments, and the revisions are described below. You can view the detailed modifications in the "Revised Manuscript with Track Changes" and "Response to Reviewers" files submitted by us.

Reviewer #1: This study explored an interesting question that does the Tobler's first law of geography apply to cyberspace by using a good example, i.e., the event of elephant migration in southwest China. The manuscript is clearly structured and well written in English. The presented data, methods, and analysis sound reasonable. I like the topic of this study and it sheds a light on the relation of “the Tobler’s first law of geography” and “network attention in cyberspace”.

However, the main issue of this study is that the conclusions are drawn only from analysis toward the event of elephant migration in southwest China. Even though it is a good case for study this topic, the conclusions are still event specific. In other word, one would question if the conclusions drawn in this study are general and solid. Some suggestions are as follows.

(1) The authors are suggested to add “a case study of *** event” in the title. And the event should also be mentioned in the abstract.

Answer：According to the comment of the reviewer, we have revised the title to "Does Tobler's first law of geography apply to internet attention? A case study of the Asian elephant northern migration event".

In addition, we have also mentioned this event in abstract.

(2) Any analysis or conclusion should be carefully expressed throughout the manuscript.

Answer：According to the comment of the reviewer, we have checked the analysis and conclusions of the whole article. Some analyses and conclusions were optimized and refined accordingly to enable a more precise presentation. Detailed revisions can be viewed in the revised manuscript.

(3) This limitation should also be emphasized in the Discussion section.

Answer：According to the comment of the reviewer, we have emphasized the limitations of the study in the discussion.

 (4) Some studies have investigated the distance-decay effect in cyberspace, which are suggested to be mentioned in the manuscript. And therefore, this manuscript should emphasize or narrow down to “network attention” instead of “cyberspace”.

* Liu, Y., Wang, F., Kang, C., Gao, Y. and Lu, Y., 2014. Analyzing Relatedness by Toponym Co-Occurrences on Web Pages. Transactions in GIS, 18(1), pp.89-107.

* Krings, G., Calabrese, F., Ratti, C., & Blondel, V. (2009). A gravity model for inter-city telephone communication networks. Journal of Statistical Mechanics: Theory and Experiment, 7003.

* Gao, S., Liu, Y., Wang, Y., & Ma, X. (2013). Discovering spatial interaction communities from mobile phone date. Transactions in GIS, 17(3), 463-481.

Answer：According to the comment of the reviewer, we focused the theme of the article on "internet attention" and revised the title to "Does Tobler's first law of geography apply to internet attention? A case study of the Asian elephant northern migration event" and revised the text accordingly.

In addition, we cite some literature investigating distance decay in cyberspace in the introduction of the paper.

(5) The authors are suggested to explain the term “network attention” earlier in the manuscript. In addition, “Internet attention” sounds better than “network attention” as “network” has a very broad meaning.

Answer：According to the comment of the reviewer, We replaced the term " network attention" with " internet attention". The first paragraph of the article explains "Internet attention".

(6) In the Discussion section, the discussion about “whether the traditional laws of geography are applicable to network space-time” should be put in the first place. The limitation of the conclusions should be emphasized (i.e., conclusions are drawn from the case study).

Answer：According to the comment of the reviewer, We discuss the limitations of Tobler's first law of geography for things in cyberspace in the first paragraph of the discussion. In addition, the limitations of this study as a case study are emphasized in the last paragraph.

(7) A small mistake in Page 11: “In the west, the strength of media publicity (X11) and geographical proximity (X11) passed the significance test.” The geographical proximity is X12 instead of X11.

Answer: According to the comment of the reviewer, we checked and calibrated the numbers of all the variables in the article.

Reviewer #2: This study aims to analyze the geographical characterize of media attention of certain event, specifically, focuses on whether the network attention following some geography law “The firs law of geography”, and further investigates the influence of geographical factors on attention. I have some comments as follow:

1. The title of this manuscript is “Does Tobler's first law of geography apply to cyberspace? An exploration based on network attention”, However, the content is little related with the topic. There is no specific experiment to verify the first law of geography, authors only conduct some conclusions from several figures. Therefore, the content does not follow the title.

Answer: As the reviewer noted, the original manuscript was not sufficiently focused on the theme of the article. Therefore, we have made some changes in the revised manuscript to further focus our research themes.

The research logic of this study is to first observe whether the closer the area to the event, the higher the internet attention is. Then, we applied the quantitative statistical method "Geo-detector" to further verify whether geographic proximity has a significant effect on internet attention at the scale of the full sample and sub-regional samples (Sub-regional samples differ in geographical proximity), given the combined effect of many factors. In the original manuscript, we did not explain the purpose of each part of the study, so we have explained the purpose of the study at the beginning of "3. Spatio-temporal characteristics" and "4. Factors Influencing the Spatio-Temporal Evolution" respectively in the revised manuscript to make the article more suitable for the topic.

In addition, we optimized the analysis of the results in the article. The focus on discussing the effect of geographic distance on Internet attention further strengthens the fit of the content of the study to the topic. The detailed changes can be found in the annotations of the revised manuscript. 

Moreover, the number of the geographic proximity variable in the original manuscript was X12, and in the revised manuscript we adjusted the number of geographic proximity to X1 to be able to emphasize the topic more.

Finally, we strengthen the discussion of the impact of geographic distance on internet attention in the "5.1 Discussion" section.

2. The term ”network attention ” is easy to have ambiguity. The network usually refers to complex network. In fact, this study just refers to news attention.

Answer: As the reviewer said, cyberspace is a complex space, and internet attention can not represent cyberspace, in fact, the object of this study should be "internet attention", so we changed the title from "Does Tobler's first law of geography apply to cyberspace? An exploration based on network attention" to "Does Tobler's first law of geography apply to internet attention? A case study of the Asian elephant northern migration event" in order to make the topic fit with the content of the study. Moreover, we explained the meaning of internet attention in the first paragraph of the article.

3.From Figure 2, the six maps are same, it can not see the difference among these figures, which makes these maps no significant. Usually, I’m more concerned about the changing value of attention over time. For example, the attention in Province A and B are 100 and 1000 respectively on week 4, then the attention on week 7 are 400 and 1100. We can see that the changing value is 300 for Province A, while the changing value is only 100 for Province B. if we only visualize the total attention, it may miss the process of change.

Answer:According to the comment of the reviewer, we have optimized the figures by adding values to be able to highlight the differences between the figures more.

4. In impact factor, the population should play an important role on attention, the more population is, the more attention should be. The population density could not reflect this phenomenon.

Answer:According to the comment of the reviewer, we changed the variable "population density" to "population", recalculated the results of the influencing factors, and made the corresponding adjustments throughout the text.

5. A conclusion should be given in the manuscript.

Answer:According to the comment of the reviewer, we have added a section "5.2 Conclusion" to explain the main conclusions of this study.

---

## [Decision Letter · Decision Letter 1]

16 Feb 2023

Does Tobler's first law of geography apply to internet attention? A case study of the Asian elephant northern migration event

PONE-D-22-25265R1

Dear Dr. QI,

We’re pleased to inform you that your manuscript has been judged scientifically suitable for publication and will be formally accepted for publication once it meets all outstanding technical requirements.

Kind regards,

Wei Tu

Academic Editor

PLOS ONE

Additional Editor Comments (optional):

Reviewers' comments:

Reviewer's Responses to Questions

**Comments to the Author**

1. If the authors have adequately addressed your comments raised in a previous round of review and you feel that this manuscript is now acceptable for publication, you may indicate that here to bypass the “Comments to the Author” section, enter your conflict of interest statement in the “Confidential to Editor” section, and submit your "Accept" recommendation.

Reviewer #1: All comments have been addressed

Reviewer #2: All comments have been addressed

2. Is the manuscript technically sound, and do the data support the conclusions?

Reviewer #1: Yes

Reviewer #2: Yes

3. Has the statistical analysis been performed appropriately and rigorously? 

Reviewer #1: Yes

Reviewer #2: Yes

4. Have the authors made all data underlying the findings in their manuscript fully available?

Reviewer #1: Yes

Reviewer #2: No

5. Is the manuscript presented in an intelligible fashion and written in standard English?

Reviewer #1: Yes

Reviewer #2: Yes

6. Review Comments to the Author

Reviewer #1: The authors have addressed your comments raised in a previous round of review and you feel that this manuscript is now acceptable for publication.

Reviewer #2: The authors have addressed all my comments, I recommend that this manuscript could be accepted for publication.

7. PLOS authors have the option to publish the peer review history of their article (what does this mean?). If published, this will include your full peer review and any attached files.

Reviewer #1: **Yes: **Kang Liu

Reviewer #2: No

---

## [Editor Report · Acceptance letter]

21 Feb 2023

PONE-D-22-25265R1 

Does Tobler's first law of geography apply to internet attention? A case study of the Asian elephant northern migration event 

Dear Dr. QI:

I'm pleased to inform you that your manuscript has been deemed suitable for publication in PLOS ONE. Congratulations! Your manuscript is now with our production department. 

Kind regards, 

on behalf of

Dr. Wei Tu 

Academic Editor

PLOS ONE